# Length and Type of Antibiotic Prophylaxis for Infection Prevention in Adults Patient in the Cardiac Surgery Intensive Care Unit: A Narrative Review

**DOI:** 10.3390/antibiotics14090934

**Published:** 2025-09-16

**Authors:** Raffaele Barbato, Benedetto Ferraresi, Massimo Chello, Alessandro Strumia, Ilaria Gagliardi, Francesco Loreni, Alessia Mattei, Giuseppe Santarpino, Massimiliano Carassiti, Francesco Grigioni, Mario Lusini

**Affiliations:** 1Unit of Cardiac Surgery, Department of Medicine, Fondazione Policlinico Universitario Campus Bio-Medico, 00128 Rome, Italy; r.barbato@policlinicocampus.it (R.B.); benedetto.ferraresi@unicampus.it (B.F.); i.gagliardi@policlinicocampus.it (I.G.); francesco.loreni@unicampus.it (F.L.); m.lusini@policlinicocampus.it (M.L.); 2Department of Anesthesia and Intensive Care, Fondazione Policlinico Universitario Campus Bio-Medico, 00128 Rome, Italy; a.strumia@policlinicocampus.it (A.S.); a.mattei@policlinicocampus.it (A.M.); m.carassiti@policlinicampus.it (M.C.); 3Department of Health Sciences, University “Magna Græcia” of Catanzaro, 88100 Catanzaro, Italy; santarpino@unicz.it; 4Unit of Cardiac Science, Department of Medicine, Fondazione Policlinico Universitario Campus Bio-Medico, 00128 Rome, Italy; f.grigioni@policlinicocampus.it

**Keywords:** antibiotic prophylaxis, cardiac surgery, surgical site infection, ICU, ECMO, heart transplant, antimicrobial resistance, multidrug-resistant organisms, perioperative infection, pharmacokinetics

## Abstract

Background: Infections following cardiac surgery are a significant cause of morbidity and mortality, particularly in intensive care units (ICUs). The role of antibiotic prophylaxis (AP) in preventing surgical site infections (SSIs) and other nosocomial infections is crucial; however, the optimal approach to agent selection, dosing, and duration remains controversial. Objective: This narrative review aims to summarise the current evidence and expert recommendations regarding the use of perioperative antibiotic prophylaxis (AP) in adults undergoing cardiac surgery, with a particular focus on intensive care settings, transplant recipients, and adult patients on extracorporeal membrane oxygenation (ECMO). Methods: A comprehensive review of recent literature was conducted, focusing on pharmacokinetic/pharmacodynamic (PK/PD) principles, microbial epidemiology, antimicrobial resistance (AMR), and practical strategies for tailored prophylaxis in high-risk populations. Results: Cefazolin remains the first-line agent for most procedures, with vancomycin or clindamycin reserved for patients who are allergic to β-lactams or who are colonised with MRSA. Redosing is recommended in cases of prolonged surgery or cardiopulmonary bypass. Evidence supports limiting prophylaxis to ≤24 h, with a potential extension to 48 h in select high-risk cases; however, continuation beyond this is discouraged due to the risk of resistance. In heart transplantation, multimodal prophylaxis against bacteria, fungi, and viruses is essential but must be tailored to the individual patient. In the ECMO setting, the current evidence does not support the routine administration of prophylaxis (AP), and therapy should be tailored based on pharmacokinetics (PK)/pharmacodynamics (PD) changes and the clinical context. A multidisciplinary, evidence-based approach to AP in cardiac surgery is essential. Prophylaxis should be patient-specific, microbiologically guided, and limited in duration to reduce the emergence of multidrug-resistant organisms. Integrating antimicrobial stewardship, non-pharmacological measures, and rigorous surveillance is crucial for optimising the prevention of infections in this vulnerable population.

## 1. Introduction

Care-related infections (CRIs) in cardiac surgery intensive care units (ICUs) pose a serious threat to public health. These result in increased mortality rates, prolonged hospital stays, and higher healthcare costs [1,2,3]. The use of antibiotic prophylaxis (AP) has attracted considerable interest and controversy, particularly regarding the type of antibiotic used and the length of the treatment period. This narrative review critically analyzes the existing literature to inform clinical practitioners and promote the safe and effective use of AP in critically ill patients [4,5,6,7,8,9,10,11]. Surgical site infections (SSIs) account for 14% to 18% of all healthcare-associated infections and are the third most frequently recorded nosocomial infections [3,4]. Several factors can increase the risk of surgical site infection, including patient-related factors such as age, nutritional status, diabetes, smoking, obesity, coexisting infections at a remote site, colonisation with a pathogenic microorganism, and an altered immune response [3,4,12]. Other factors include operative procedure-related factors such as the duration of surgical scrub skin antisepsis, preoperative shaving, preoperative skin preparation, duration of surgery, antibiotic prophylaxis, operating room ventilation, inadequate sterilisation of instruments, presence of foreign material at the surgical site, use of surgical drains, and surgical technique [3,4,13,14]. Different surgical sites pose different levels of risk. The risk of a surgical site infection also depends on the type of procedure, varying according to whether it is clean, clean-contaminated, contaminated, or dirty-infected [3,4,13]. Improvements in operating room ventilation and sterilisation methods, surgical barriers and surgical techniques, and the use of topical, oral, and intravenous perioperative antibiotics have all played an important role in reducing the incidence of SSIs [3,4,12]. Perioperative surgical antimicrobial prophylaxis is recommended for surgical procedures involving a high risk of postoperative wound infection, procedures involving the implantation of foreign material, and low-risk procedures in which the development of an infection would have disastrous consequences [3,4,15]. Prophylactic antimicrobial agents should be bactericidal, non-toxic, inexpensive, and demonstrate in vitro activity against the common organisms that cause postoperative wound infections following specific surgical procedures [3,4,15]. They should be administered 30–60 min before surgery [4,15]. This narrative review summarises the existing literature on AP in cardiac surgery intensive care, including preoperative prophylaxis and considerations for transplant and mechanically supported patients [1,3,5,13,16,17,18,19].

## 2. Antimicrobial Prophylaxis in Cardiac Surgery

### 2.1. Pharmacokinetic/Pharmacodynamic Rationale and Antibiotic Selection

#### 2.1.1. PK/PD Principles Applied to Cardiac Surgery

The aim of AP is to ensure that bactericidal drug concentrations are present in the serum and tissues throughout the entire period of microbial exposure [15,20]. For time-dependent antibiotics (e.g., β-lactams), it is crucial to maintain a free (unbound) concentration above the minimum inhibitory concentration (MIC) for a substantial proportion of the dosing intervals. Conversely, for concentration-dependent agents (e.g., aminoglycosides and fluoroquinolones), the main determinant of efficacy is achieving sufficiently high plasma concentrations, followed by drug-free intervals. In cardiac surgery, cardiopulmonary bypass (CPB) alters the volume of distribution and half-life of drugs. This sometimes necessitates larger doses or re-dosing during surgery to maintain effective drug concentrations [21]. Furthermore, specific conditions, such as obesity, hypoalbuminaemia, and renal impairment, necessitate targeted dose adjustments [21].

#### 2.1.2. Pathogen Spectrum and First-Line Agents

In cardiac surgery, surgical site infections (SSIs) are most likely caused by *Staphylococcus* spp. (including both methicillin-susceptible and methicillin-resistant S. aureus) and *Streptococcus* spp. Although Gram-negative organisms may be relevant in high-risk scenarios involving prosthetic implants, reoperations, or prolonged hospitalisation [12,15,21], first-generation cephalosporins, particularly cefazolin, are considered the gold standard for prophylaxis in this setting due to their reliable activity against the most common Gram-positive cocci, favourable safety profile, and cost-effectiveness [15,22]. Studies and guidelines indicate that appropriately dosed cefazolin (e.g., 2 g for patients weighing >80 kg and 3 g for patients weighing >120 kg) yields protective tissue levels when administered at the correct time.

#### 2.1.3. Alternatives in Case of β-Lactam Allergy or High MRSA Prevalence

Alternative agents, such as vancomycin or clindamycin, are recommended for patients with documented severe allergy to β-lactams or in geographic areas with a high prevalence of methicillin-resistant S. aureus (MRSA) [3,15]. Although vancomycin is effective against MRSA, it provides inadequate coverage against Gram-negative bacteria. Its routine use without a specific indication may lead to the development of resistant strains; therefore, it should only be used in patients with a clear indication (e.g., severe β-lactam allergy or known MRSA carriage) [12,15]. Clindamycin can be used in patients who are allergic to β-lactams and offers good coverage against Gram-positive bacteria. However, it carries a risk of Clostridioides difficile infection and must, therefore, be used cautiously in accordance with local epidemiology [22,23].

#### 2.1.4. Additional Coverage for Gram-Negative Organisms

For individuals at high risk of Gram-negative infections (e.g., those undergoing reoperation on the sternum with prosthetic implants or with hospital-acquired colonisation), adding a single dose of an aminoglycoside (e.g., gentamicin), aztreonam, or a fluoroquinolone may be beneficial, provided the potential benefits outweigh the risks of toxicity and selection pressure for resistance [4,12,15]. In the UK and Ireland, a regimen combining flucloxacillin with a single pre-incision dose of gentamicin is used to provide balanced coverage while minimising the impact on the microbiota and the risk of *C. difficile* infection [5,13].

#### 2.1.5. Adjustments in Specific Subgroups

##### Obesity

Patients with a high BMI require higher doses of cefazolin to maintain effective tissue concentrations (e.g., 2 g for patients weighing >80 kg and 3 g for patients weighing >120 kg). Some studies have reported significant reductions in infection rates when the dose is increased for obese patient groups (for example, a reduction from 16% to 5.6% in certain settings).

##### Cardiopulmonary Bypass (CPB) and Blood Loss

In cases involving CPB or substantial intraoperative blood loss (>1–2 litres), re-dosing should be considered intraoperatively, based on the drug’s half-life and the timing of the initial dose, rather than solely on incision time, in order to sustain protective levels.

##### Renal/Hepatic Impairment

The dosage should be adjusted according to renal clearance while ensuring continued protective efficacy. In this case, it should be noted that CPB-related haemodilution may temporarily affect the actual clearance.

#### 2.1.6. Prophylaxis Timing

##### General Timing Guidelines

The initial antibiotic dose must be administered such that effective concentrations are present in the tissues at the time of the incision. For short half-life β-lactams (e.g., cefazolin and cefoxitin), the dose should ideally be administered within 30 min and no later than 60 min before the incision to maximise tissue levels. For vancomycin and certain fluoroquinolones, infusion should start up to 120 min before the incision to allow time for completion [13,23].

##### Evidence on Administration Intervals

Analyses of comparative studies show that administering antibiotics at any time within 120 min before the incision (e.g., 120–60 min vs. 60–0 min, or 60–30 min vs. 30–0 min) has no significant impact on Surgical Site Infection (SSI) rates [12,13,14,15,16,24]. In practice, the timing of administration should be planned according to the specific antibiotic: for β-lactams, the dose should be administered as close as possible to the incision (within 60 min), whereas for vancomycin, it should be administered up to 120 min prior to the incision [4,9,16].

##### Intraoperative Re-Dosing

An intraoperative re-dose should be administered if the surgery exceeds two half-lives of the antibiotic or if significant blood loss occurs. This should be calculated from the time of the initial administration rather than from the time of incision alone, in order to maintain effective levels until wound closure. This is particularly important during prolonged cardiopulmonary bypass (CPB) or lengthy procedures [9,21].

##### Organizational Aspects and Compliance

To ensure adherence to timing, it is helpful for anaesthetists, surgeons, and pharmacy staff to share protocols, perioperative alert systems, and checklists. Regular staff training and audits can also help minimise dosing errors [5,18].

#### 2.1.7. Duration of Prophylaxis

##### Overall Duration Based on Evidence

A single pre-incision dose is effective for low-risk procedures and avoids unnecessary prolongation. However, for high-risk procedures, such as cardiac surgery, which carries a high risk of severe complications, such as mediastinitis, this alone is generally considered insufficient [2,23].

≤24 h: This is the standard for most cardiac surgery procedures. Meta-analyses and randomised controlled trials indicate that it is as effective as longer regimens in preventing surgical site infections (SSIs), and that it reduces the development of resistance and the occurrence of antibiotic-related complications [10,16,17].

24–48 h: Reserved for selected cases (e.g., prosthetic implants, reoperations, and complex sternotomies), provided local guidelines and microbiological surveillance support this extension. Many societies (e.g., the Society of Thoracic Surgeons) permit an extension to 48 h for high-risk cardiac surgery patients, but emphasise that there is no benefit beyond this duration [10,13,25].

>48 h: Not recommended, as it does not improve SSI prevention, but increases the selection of resistant strains (e.g., MRSA and multidrug-resistant Gram-negatives), raises the incidence of C. difficile infection, and fosters nosocomial complications [9,26,27].

##### Evidence Comparing 24 vs. 48 h

Meta-analyses and randomised trials show that 24-h prophylaxis is as effective as 48-h regimens at preventing surgical site SSIs in cardiac surgery. However, a longer duration favours resistant strain selection without providing any additional benefits. While some studies suggest that 24 h may reduce deep infections compared to a single dose, there is no strong evidence that extending the duration to 48 h offers general advantages. Thus, any extension beyond 24 h should be assessed on a case-by-case basis and justified by specific risk factors [10,13,17].

##### Management in the Presence of Intrathoracic Drains

Historically, prophylaxis was stopped and catheters were removed to ‘protect’ any potential entry points. However, there is consistent but low-quality evidence that prolonged prophylaxis in the presence of drains has no effect on surgical site infection (SSI) rates compared to perioperative prophylaxis alone. Similarly, studies comparing early drain removal (days 1–5) with late removal (day 6 onwards or volume-based) reveal no significant difference in SSI rates. Therefore, the conditional recommendation based on the presence of drains is not to continue prophylaxis beyond wound closure. Instead, drains should be removed according to specific clinical criteria (e.g., bleeding and signs of local inflammation) rather than fixed antibiotic duration protocols. Extending prophylaxis due to the presence of drains exposes patients to resistant strains, superinfections, and drug toxicity without providing additional protection against SSIs. Therefore, decisions regarding duration must be evidence-based rather than driven by historical routines [10,16,25,28].

#### 2.1.8. Multidisciplinary Integration and Complementary Measures

##### Antimicrobial Stewardship and Epidemiological Monitoring

The choice and duration of prophylaxis should be tailored to local resistance epidemiology (e.g., MRSA prevalence, multidrug-resistant Gram-negative bacteria, and *C. difficile* incidence) [3,5,15]. A continuous surveillance programme with regular feedback to surgical and infectious disease teams and protocol updates helps contain the onset of resistance [5,23,26].

##### Complementary Non-Pharmacological Measures

The choice and duration of prophylaxis should be tailored to the local epidemiology of resistance (e.g., prevalence of MRSA, multidrug-resistant Gram-negative bacteria, and incidence of C. difficile) [3,5,15]. A continuous surveillance programme involving regular feedback to surgical and infectious disease teams, as well as protocol updates, helps to contain the onset of resistance [5,23,26].

##### Complementary Non-Pharmacological Measures

In addition to antibiotic prophylaxis, strict surgical asepsis, skin preparation with antiseptics, and infection prevention bundles (e.g., hand hygiene, vascular catheter care, and ventilator-associated pneumonia prevention) are essential. Nasal decolonisation with mupirocin in *S. aureus* carriers can reduce staphylococcal SSIs by up to 45%, which highlights the importance of preoperative screening for multidrug-resistant organism colonisation, particularly in higher-risk patients (e.g., those transferred from other institutions or with prolonged hospital stays). In addition to nasal decolonisation, additional prophylactic vancomycin therapy is useful in MRSA-positive patients [2,4]. Antiseptic irrigation, antimicrobial-impregnated sutures, and specialised dressings may be considered in selected protocols, but they do not replace standard systemic prophylaxis. In addition to placing vancomycin powder in surgical incisions, antimicrobial-coated sutures have recently been developed as a possible adjunct to reduce the rate of SSIs. However, clinical trials have shown variable efficacy of antimicrobial sutures, making it difficult to recommend their use [12,28].

##### Role of the Multidisciplinary Team

Optimising prophylaxis requires close collaboration between surgeons, anaesthetists, infectious disease specialists, and pharmacists. Pharmacists assist with dosing calculations and timing, infectious disease experts provide regimen recommendations based on microbiological data, and surgeons assess intraoperative factors, such as duration, blood loss, and reoperations. Anaesthetists ensure correct administration and re-dosing. Periodic audits of protocol adherence help identify gaps and areas for improvement [13,18].

#### 2.1.9. Practical Summary Recommendations

The appropriate selection and timing of AP are critical for reducing the risk of SSIs, particularly in cardiac surgery [1,2,4]. First-line prophylaxis typically involves the use of cefazolin, which is administered according to the patient’s weight and taking into account CPB pharmacokinetics [1,21]. For most patients, the appropriate dose is 2–3 g, adjusted for body mass index.

For patients with a documented severe beta-lactam allergy or known or suspected MRSA colonisation, alternative agents such as vancomycin or clindamycin are recommended [4,15]. In selected high-risk cases, particularly when there is a concern about Gram-negative pathogens, a single preoperative dose of an aminoglycoside or another targeted agent may be administered. However, the risks of nephrotoxicity and the potential for promoting antimicrobial resistance must be carefully weighed [12,15]. Timing is essential. For β-lactam antibiotics with a short half-life, administration should occur 30–60 min before the surgical incision. In contrast, agents such as vancomycin or fluoroquinolones, which require longer infusion times, should be initiated 60 to 120 min before the start of surgery. For longer procedures, particularly those exceeding two drug half-lives, involving substantial intraoperative blood loss, or requiring cardiopulmonary bypass, repeat dosing may be warranted to maintain adequate tissue levels. Timing should be calculated from the initial administration [9,16,21].

In terms of duration, a perioperative course limited to ≤24 h is usually sufficient for standard cardiac procedures. In complex cases involving reoperations, prosthetics, or high-risk patients, a 48-h extension may be justified [5,9,15]. However, there is no demonstrated benefit beyond 48 h, and prolonged use increases the risk of resistance and side effects [3,12,15]. Importantly, the mere presence of surgical drains should not dictate the continuation of antibiotic prophylaxis; drain removal should be based on clinical judgement and drainage volume rather than the duration of antibiotic administration [9,12].

Lastly, AP must be part of a broader, integrated infection prevention strategy. This includes adherence to aseptic techniques, implementation of decolonisation protocols (such as nasal mupirocin for MRSA), and compliance with infection prevention bundles [2,12,14]. Multidisciplinary collaboration, regular protocol audits, and adjustments based on local microbiological surveillance are essential to ensure effective prophylaxis while minimising unnecessary antimicrobial exposure [3,5].

## 3. Multidrug Antimicrobial Resistance: Mechanisms, Clinical Impact, and Intervention Strategies

Antimicrobial resistance (AMR) poses a major global public health threat and has direct implications for outcomes in cardiac surgery ICUs. Multidrug resistance (MDR) is defined as non-susceptibility to at least one agent in three or more antimicrobial classes. The main causes of this phenomenon are the indiscriminate use of antibiotics in human and veterinary medicine, the transfer of genes between bacterial species, and the chronic shortage of new antibacterial compounds in development.

In 2017, the World Health Organization (WHO) published a prioritised list of antibiotic-resistant ‘priority pathogens’, categorising them as critical, high, or medium priority based on the urgency of developing new treatments. Critical-priority organisms include carbapenem-resistant *Enterobacteriaceae*, Pseudomonas aeruginosa, and Acinetobacter baumannii, which cause severe nosocomial pneumonia and bloodstream infections in hospitalised patients [27].

### 3.1. Clinical Impact in Cardiac Surgery Intensive Care

Intensive care units (ICUs) are epicentres for the dissemination of multidrug-resistant (MDR) organisms. Patients undergoing complex cardiac procedures are often critically ill and exposed to invasive devices (ventilators, central lines, and chest drains). They also receive broad-spectrum antibiotics, all of which increase susceptibility to MDR infections. Despite the implementation of rigorous infection control measures, including hand hygiene, active surveillance, and contact isolation, the prevalence of MDR colonisation and infection in ICUs continues to rise [29].

A recent meta-analysis reported an incidence of new MDR intestinal colonisation of 1.7 per 1000 patient-days in ICU settings, with a median acquisition time ranging from four to 26 days. This risk increases linearly for up to 30 days and varies by pathogen, with a higher risk for non-Pseudomonas Gram-negatives than for *P. aeruginosa* [29].

For patients undergoing cardiac surgery, the development of ventilator-associated pneumonia, bloodstream infection, or sternal wound infection caused by MDR organisms can lead to prolonged ICU stays, higher costs, and increased mortality [1,2,28,29].

### 3.2. Selective Decontamination and Antibiotic Cycling

Nasal decolonisation with topical mupirocin, often combined with chlorhexidine bathing, has been shown to reduce MRSA-related SSIs in cardiac surgery [30]. However, the ecological safety of this approach in regions with high levels of ESBL or carbapenemase-producing bacteria requires further evaluation [31,32,33]. Periodic rotation of empirical antibiotic therapy aims to limit resistance; however, there are insufficient long-term clinical data in the cardiac surgery setting to confirm its effectiveness [29,34].

### 3.3. Innovative Therapeutics and Combination Approaches

In response to the increasing threat posed by multidrug-resistant (MDR) pathogens, such as MRSA, CRE, VRE, MDR Pseudomonas aeruginosa, and MDR *E. coli*, a variety of novel and repurposed therapeutic strategies are being investigated [31,32]. One promising approach is the use of synergistic combinations of drugs or plant-derived compounds, such as ethyl gallate combined with tetracycline or fusidic acid, which have shown enhanced activity against MRSA [31].

New combinations of β-lactams and β-lactamase inhibitors, such as ceftolozane/tazobactam and ceftazidime/avibactam, have been shown to be effective against resistant strains of *P. aeruginosa* [29,31].

The development of next-generation glycopeptides with enhanced potency is also progressing, particularly for treating vancomycin-resistant Enterococcus (VRE) [31,32]. Other emerging strategies include the use of antimicrobial polymeric biomaterials and nanoparticle-based drug delivery systems, which offer targeted and sustained antimicrobial release, particularly in biofilm environments [12,31]. Furthermore, botanical extracts and phytochemicals with broad-spectrum activity are being explored, although most of the evidence remains at the preclinical stage [31].

Other studies have used lytic bacteriophages that have been carefully selected through genomic profiling. For example, the phage phiLLS has been shown to be specific to multidrug-resistant (MDR) *E. coli* strains. Finally, certain probiotic strains, such as Lactobacillus plantarum Y3, can prevent MDR organisms from forming biofilms on indwelling medical devices [31].

While some of these approaches, including bacteriophage therapy and probiotics, are still experimental, they show promise as adjuncts in high-risk cardiac surgical populations [29,31,33,35].

### 3.4. Preoperative Colonization and Surgical Risk

Preoperative multidrug-resistant (MDR) colonisation presents additional challenges in surgical settings, especially for major procedures, such as cardiac surgery. Recently, efforts have been made to establish guidelines for managing preoperative MDR colonisation. However, the authors of these guidelines acknowledged that the strength of the recommendations and quality of the supporting evidence were generally low [36]. Although some studies have identified an increased risk of postoperative infections in cardiac surgery patients colonised by MDR pathogens preoperatively, a recent retrospective study of patients undergoing bypass grafting or valve replacement found no statistically significant differences in postoperative outcomes (bacteraemia, pneumonia, surgical site infection, ICU stay, and mortality) between colonised and non-colonised cohorts. Nevertheless, over half of the infections in the colonised group were caused by the same preoperative colonising pathogens. Meticulous perioperative management, including contact isolation, targeted prophylaxis, and eradication when feasible, may mitigate the risk of postoperative MDR infections [34,36,37].

### 3.5. Future Perspectives

The ongoing evolution of AMR requires a differentiated response, including active surveillance, rigorous antimicrobial stewardship, infection prevention, and accelerated development of antibiotics.

Although they require more time, public–private partnerships are pivotal in replenishing the antibiotic pipeline.

Continuous education for healthcare personnel and patient engagement in the rational use of antibiotics are essential to slow the emergence of resistance and preserve the effectiveness of existing therapies.

## 4. Antibiotic Prophylaxis for Patients Receiving Heart Transplants

Heart transplantation is one of the most advanced life-saving therapies for patients with end-stage heart failure that cannot be treated with medications. However, the success of this procedure is jeopardised by a number of infectious complications, primarily due to the immunosuppression required to prevent graft rejection. In this context, antimicrobial prophylaxis plays a crucial role in preventing infections and significantly influences postoperative morbidity and mortality [19,25].

This review provides a detailed overview of current antibacterial, antifungal, and antiviral prophylaxis strategies for heart transplant recipients, based on an analysis of the available data. The analysis is based on retrospective evidence from large observational cohorts and outlines the treatment protocols implemented, as well as the incidence and temporal evolution of post-transplant infections [19,28].

### 4.1. Postoperative Infections: Epidemiology and Clinical Impact

According to one of the analysed studies, postoperative infection is one of the most common complications of heart transplantation, with an in-hospital incidence rate of 35.3% [27]. Most infections occur with the use of central venous catheters, mechanical ventilation, and parenteral nutrition. The cumulative incidence rate has increased over time, probably reflecting improved diagnostic capacity rather than an actual increase in vulnerability [25].

The most common infection sites were the respiratory tract (33%), urinary tract (13.5%), bloodstream (12.4%), and abdominal cavity (9.5%) [27]. The predominant pathogens were Gram-negative bacteria (*Enterobacteriaceae* and Pseudomonas aeruginosa), followed by Gram-positive bacteria (including *Staphylococcus aureus* and MRSA). Although less frequent, fungal and viral infections pose a significant threat to prognosis, particularly in immunocompromised patients [19].

### 4.2. Antibacterial Prophylaxis: Rational Use and Limitations

Despite the lack of randomised controlled trials to define a universally optimal protocol, the administration of perioperative antibiotics (AP) is a cornerstone of practice in all transplant centres. The most commonly used regimens include first-generation cephalosporins (e.g., cefazolin) for patients who are not allergic, and a combination of vancomycin and a fluoroquinolone for patients who are allergic or colonised with MRSA [28].

For hospitalised patients with an LVAD, the combined use of piperacillin-tazobactam and vancomycin is a more aggressive approach, often justified by the presence of multidrug-resistant nosocomial pathogens. However, data show that prolonged AP beyond 48 h post-operatively does not reduce infection rates, but may contribute to the development of resistant strains. Therefore, discontinuing prophylaxis within 48 h appears to be an effective and simple strategy to avoid inappropriate use of antimicrobial agents [28].

### 4.3. Antifungal Prophylaxis: Targeted Selection and Risk Containment

Approximately half of transplant centres employ perioperative antifungal prophylaxis, often based on empirical decisions rather than strong evidence [25]. Early fungal infections are relatively rare and typically appear weeks or months after transplantation. The routine use of systemic antifungal prophylaxis, particularly azoles, in all patients may increase the risk of selecting resistant fungal strains [25].

Recent studies have confirmed the effectiveness of risk-based antifungal prophylaxis in selected high-risk patients, such as those with active preoperative infections, prolonged hospitalisation, or parenteral nutrition. A study at Stanford University found that the combination of inhaled amphotericin B and oral itraconazole for three months after transplantation significantly reduced *Aspergillus* spp. infections, which are associated with high mortality. Additionally, HEPA-filtered respiratory masks have been found to be effective non-pharmacological measures.

### 4.4. Antiviral Prophylaxis: The Central Role of CMV

Among all viral infections, cytomegalovirus (CMV) poses the greatest threat to heart transplantation. Although prophylaxis with valganciclovir, which is usually administered for 6–12 months, has been shown to reduce the incidence of CMV infection, it does not completely eradicate it [19]. One study demonstrated that extending the prophylaxis period from six to 12 months did not reduce the frequency of infection; rather, it merely delayed its onset to a period of lower immunosuppressive intensity, which could potentially reduce clinical severity [19].

However, some questions remain unanswered. The minimum effective duration of prophylaxis is unclear, as is the question of whether post-prophylaxis monitoring alone can prevent severe disease. Decisions regarding the duration of prophylaxis should probably be individualised based on immunological risk, CMV serostatus mismatch, and comorbid conditions [19].

### 4.5. Opportunistic Infections and Integrated Strategies

Opportunistic infections caused by *Pneumocystis jirovecii*, *Toxoplasma gondii*, and *Nocardia* spp. have significantly decreased owing to the systematic use of targeted prophylaxis involving TMP-SMX or atovaquone. TMP-SMX remains the primary agent for prophylaxis against these infections, while atovaquone is only used as an alternative in cases of TMP-SMX intolerance. [25]

Similarly, the reduction in invasive fungal infections and the near-total absence of Mycobacterium tuberculosis or Legionella species confirm the success of modern prevention protocols [25]. However, the occurrence of rare infections (e.g., reactivation of Chagas disease or Strongyloides infection) emphasises the importance of targeted epidemiological screening, particularly for patients from endemic regions [25].

Organisational aspects and practical recommendations

From a clinical point of view, it is important to highlight the need for prophylaxis strategies tailored to the individual rather than overly broad or uniform protocols. The variability among centres, some of which use up to four drugs in combination for surgical prophylaxis, reflects the lack of consensus and highlights the need for standardised, evidence-based guidelines [28]. Shared protocols must be developed based on local microbiological data (antibiograms), patient risk factors, and epidemiological contexts [25].

Interdisciplinary collaboration is another crucial element. Early involvement of infectious disease specialists in defining prophylaxis strategies and managing complications has been shown to lead to better clinical outcomes. Similarly, the use of rapid molecular diagnostics enables more precise and timely infection management.


**To sum up**


Antimicrobial prophylaxis in heart transplantation is a vital yet complex strategy that requires a balance between preventive efficacy, toxicity, cost-effectiveness, and risk of antimicrobial resistance. Current evidence suggests that a personalised, multimodal approach involving targeted prophylaxis against bacteria, viruses, and fungi in high-risk individuals, coupled with rigorous microbiological surveillance, is the best option.

Prospective multicentre studies are needed to evaluate the cost-benefit ratio of different prophylactic regimens and determine the optimal duration for specific patient subgroups. Meanwhile, the most rational way to mitigate the burden of infection in heart transplant recipients is to integrate existing evidence and clinical expertise with local adaptation.

## 5. ECMO

### 5.1. Introduction

Extracorporeal membrane oxygenation (ECMO) is an increasingly utilised supportive therapy for critically ill patients with refractory respiratory or cardiac failure. Despite its life-saving potential, ECMO is associated with a significant risk of nosocomial infection (NIs), which contributes to increased morbidity and mortality. The estimated incidence of sepsis following cannulation is up to 42%, with nosocomial infections affecting approximately 26% of patients undergoing ECMO (reported ranges from 1% to 93%) [11,22]. These infections are associated with a mortality risk that is almost double that of patients without infections, with an odds ratio of 1.91 [11].

### 5.2. Pathophysiology of Infection During ECMO

The pathogenesis of infection in ECMO patients is multifactorial. Contact between blood and foreign surfaces within the ECMO circuit induces a systemic inflammatory response, leading to the activation of the coagulation cascade, complement system, and leukocytes, as well as the release of pro-inflammatory cytokines. This environment contributes to immunoparalysis, making patients more susceptible to secondary infection. Furthermore, prolonged ECMO duration, particularly beyond 14 days, is strongly correlated with increased infection rates [18].

### 5.3. Microbial Epidemiology

The most commonly isolated organisms include coagulase-negative staphylococci, *Enterococcus* spp., *Pseudomonas aeruginosa*, *Klebsiella pneumoniae*, and *Candida* spp. Bloodstream infections (BSI) and ventilator-associated pneumonia (VAP) are the most frequent nosocomial complications. The prevalence of resistant pathogens, including multidrug-resistant organisms (MDROs), is also increasing, further complicating treatment and infection control [11,21].

### 5.4. Prophylactic Antibiotic Strategies

Despite its widespread use, the efficacy of prophylactic antibiotic therapy during extracorporeal membrane oxygenation (ECMO) remains controversial. Several observational studies and meta-analyses have failed to demonstrate a significant reduction in 30-day mortality rates [18,34]. However, some evidence suggests a modest reduction in infection rates, with an NNT of 40 to prevent a single nosocomial infection. However, these data must be interpreted with caution due to the retrospective nature and heterogeneity of the included studies [11,34].

International guidelines, including those from the Extracorporeal Life Support Organisation (ELSO), do not recommend routine antibiotic prophylaxis (AP) in ECMO patients, except for standard perioperative indications [34]. Nevertheless, surveys have revealed that up to 74% of centres still adopt empirical prophylaxis, particularly for high-risk groups, such as patients undergoing open-chest procedures.

### 5.5. Diagnostic Challenges

ECMO alters the host’s immune response, which complicates the interpretation of classic infection markers such as fever, leukocytosis, C-reactive protein (CRP), and procalcitonin (PCT). Due to extracorporeal heat exchange, febrile responses are often attenuated. Furthermore, routine laboratory markers have shown low specificity in this context [38]. Blood and respiratory cultures remain critical for diagnosis; however, their interpretation can be complicated by colonisation or contamination [11]. The overuse of routine surveillance cultures is not currently supported due to their low yield and the potential for false positives.

### 5.6. Antifungal Prophylaxis and Risk

Antifungal prophylaxis has not been shown to be clearly beneficial for ECMO patients without haematological malignancies or neutropenia [34]. The risk of Candida infections appears to be higher in premature neonates and certain adult populations [38]. The use of azoles, such as fluconazole, should be considered alongside potential pharmacokinetic alterations from ECMO circuits and the risk of QT prolongation [34].

### 5.7. Pharmacokinetics and Dosing Considerations

ECMO can significantly alter the pharmacokinetics (PK) and pharmacodynamics (PD) of the administered antibiotics. Changes in the volume of distribution, drug sequestration by circuit components, and the administration of concurrent renal replacement therapy can all impair drug efficacy [34,38]. Therapeutic drug monitoring (TDM) is therefore strongly recommended, particularly for time-dependent or narrow therapeutic index agents such as vancomycin and teicoplanin. Inadequate dosing can result in treatment failure, whereas overdose can increase the risk of nephrotoxicity.

### 5.8. Clinical Outcomes and Cost Considerations

While prophylaxis may marginally reduce the incidence of infection, it does not appear to significantly affect survival [27]. Furthermore, the routine use of antibiotics can lead to the emergence of MDROs, increased costs, and adverse drug reactions, such as Clostridioides difficile colitis [34]. Economic analyses suggest that a targeted, on-demand approach based on clinical criteria and microbiological data may be more cost-effective [34,36,37,38,39,40].

## 6. Conclusions

A comprehensive approach is needed to prevent infections during cardiac surgery, heart transplantation, and extracorporeal membrane oxygenation (ECMO). Although AP remains a vital component of prevention strategies, it must be employed judiciously. Effective strategies for reducing SSIs in cardiac surgery without increasing the risk of resistance include using cefazolin as a first-line agent [1,2], adjusting dosage based on body weight and the presence of cardiopulmonary bypass [21], and limiting the duration of treatment to 24–48 h [5,9,15]. Using it for more than 48 h offers no additional benefits and promotes the emergence of resistant strains [10,11,16].

In heart transplantation, where susceptibility to infection is increased by immunosuppression, prophylaxis against bacteria, fungi, and viruses is necessary [19,25]. However, extending prophylaxis beyond the recommended timeframe unnecessarily is still associated with an increased risk of toxicity and the development of multidrug-resistant strains [28]. Prophylaxis should be guided by the individual patient’s risk factors, local microbial flora, and epidemiological data [25]. Particular attention should be given to special populations, such as patients with LVADs, patients with MDR colonisation, and those from endemic areas [25,27].

Current evidence does not support the routine use of AP in ECMO support, except for well-defined perioperative indications [34]. Pharmacokinetic alterations induced by the ECMO circuit, the difficult interpretation of inflammatory markers, and high variability in clinical practice necessitate highly individualised management supported by therapeutic drug monitoring and interdisciplinary collaboration. Furthermore, indiscriminate use of AP has little impact on mortality but increases costs and complications associated with antimicrobial resistance [34,36].

The following are confirmed as crucially important across the board: a personalised, risk-based approach and the integration of non-pharmacological interventions [6,9,23,26,29].

The Antibiotic Prophylaxis Decision-Making Process

The selection of perioperative antibiotic prophylaxis for cardiac surgery is based on a structured, patient-centred algorithm that takes into account individual risk factors and local epidemiology.

Surgical indication:

Antibiotic prophylaxis is recommended for all patients undergoing cardiac surgery due to the high risk of surgical site infections (SSIs) and potential severe complications (e.g., mediastinitis).

Assessment of β-lactam allergy or MRSA colonisation:

If the patient has a documented severe β-lactam allergy or is colonised with methicillin-resistant *Staphylococcus aureus* (MRSA), alternative agents such as vancomycin or clindamycin should be used. In all other cases, cefazolin is the preferred first-line agent due to its efficacy, safety, and cost-effectiveness.

Risk of Gram-Negative Infection

Consider a single pre-incision dose of an aminoglycoside (e.g., gentamicin), aztreonam, or a fluoroquinolone to broaden coverage for patients at high risk of Gram-negative infection, such as those undergoing reoperations (e.g., redo sternotomy), with documented preoperative colonisation by Gram-negative organisms (especially multidrug-resistant/extended-spectrum beta-lactamase/carbapenem-resistant *Enterobacteriaceae* or *Pseudomonas*), or with prolonged preoperative hospitalisation/ICU stay [39]. Weigh the potential toxicity and local resistance ecology.

Special Population Adjustments:

Obesity: Dose adjustment of cefazolin is required to achieve therapeutic tissue levels (2 g if body weight is >80 kg; 3 g if body weight is >120 kg).

Cardiopulmonary bypass (CPB) or major blood loss: Intraoperative re-dosing may be necessary based on pharmacokinetic data.

Renal or hepatic impairment: Dosage should be modified according to clearance capacity, particularly during prolonged procedures.

Timing of administration:

β-lactams (e.g., cefazolin): Administer 30–60 min before the surgical incision is made.

Vancomycin or fluoroquinolones: Infusion is initiated up to 120 min before the incision to allow for adequate tissue penetration.

Duration of prophylaxis:

For most procedures, prophylaxis should not exceed 24 h after surgery.

In selected high-risk cases, extending the duration to 48 h may be justified.

However, prophylaxis beyond 48 h is strongly discouraged, since it does not reduce infection rates but significantly increases the risk of antimicrobial resistance and complications.

Presence of Surgical Drains

The presence of surgical drains does not justify extending antibiotic prophylaxis. Decisions regarding drain management should be based on clinical factors rather than the duration of antibiotic therapy.

Literature search: A literature search was performed using PubMed and Google Scholar to identify relevant publications from January 2000 to June 2025. The search was limited to articles published in English and Italian. Eligible studies included original research articles, systematic and narrative reviews, and clinical guidelines published in peer-reviewed journals. Articles were excluded if they were irrelevant to the research topic, duplicates, or lacked full-text availability.

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
