# Peer review of "Length and Type of Antibiotic Prophylaxis for Infection Prevention in Adults Patient in the Cardiac Surgery Intensive Care Unit: A Narrative Review"

_antibiotics, 2025, doi:10.3390/antibiotics14090934_

Round 1

Reviewer 1 Report

Comments and Suggestions for Authors

The authors in the narrative review, titled as "Length and type of antibiotic prophylaxis for infection prevention in the cardiac surgery intensive care unit: a narrative review", have utilised the peer reviewed literature to synthesize current evidence on antibiotic prophylaxis in cardiac surgery ICU settings. However, there are a few, minor formatting issues and a factual date error that need attention.  Following are the suggestions:

WHO Priority Pathogens List Date Error

  • Current text: "In 2021, the World Health Organization (WHO) published a prioritised list of antibiotic-resistant 'priority pathogens'"
  • Correction needed: Change to "In 2017, the World Health Organization (WHO) published..." or reference the 2024 update if using more recent data

Complete Citation Formatting

  • Fill in all missing DOI information currently marked as "??"

Figure/Diagram Addition

  • Consider adding a flowchart for antibiotic selection decision-making

Strengthen Methodology Section

  • Add a brief methodology paragraph describing literature search strategy
  • Specify databases searched, date ranges, and inclusion/exclusion criteria

Expand on Emerging Resistance Patterns

  • Consider adding more recent data on ESBL and CRE

Author Response

All comments have been resolved and a new article has been attached.

Reviewer 2 Report

Comments and Suggestions for Authors

The manuscript addresses important clinical issues, and the authors’ attempt to gather and organize current data on infection prophylaxis in cardiac surgery is commendable. The topic is highly relevant, both from the perspective of clinical practice and antimicrobial resistance control strategies. However, in its current form, the manuscript is not suitable for publication and requires substantial revisions.

The title and stated objectives do not accurately reflect the actual scope of the paper. The title and the “Objectives” section should be revised to clearly indicate the focus of the article. It should also be explicitly stated that the paper concerns adult patients.

In many sections, the information presented is imprecise, overly general, or inconsistent. For example, subsection 1.8.1 discusses the effectiveness of Staphylococcus aureus decolonization, whereas subsection 1.9 refers only to MRSA-colonized patients. It is unclear whether the recommendations pertain to both MSSA and MRSA or to MRSA alone. The authors should clearly specify the current guidelines and evidence level for screening and decolonization, and present the recommended decolonization protocol. Similarly, the statement: “Antiseptic irrigations, antimicrobial-impregnated sutures and specialised dressings may be considered in selected protocols” is too vague. It is unclear what types of irrigations or “specialised dressings” are being referred to, in which clinical settings they are recommended, and whether there is evidence supporting their efficacy in preventing surgical site infections (SSI). Such generalizations require clarification. In their current form, these fragments raise more questions than they provide practical guidance, thereby weakening the value of this review for clinicians.

Subsection 1.9 and the “To sum up” section largely repeat content already presented in earlier parts of Chapter 1. They do not introduce any significant new information, and the duplication of content related to antibiotic selection, dosing, timing, redosing, and duration of prophylaxis diminishes the clarity of the structure. These sections should be removed.

Although Chapter 2 addresses the important topic of antimicrobial resistance, it appears poorly aligned with the main objectives of the manuscript and differs in style and scope from the rest of the paper. Much of the content is general, selective, or speculative, and only marginally related to prophylaxis in cardiac surgery. Moreover, the definition of MDR provided by the authors is incorrect. The discussion on SOD and SDD strategies is also unclear. Although the authors mention these approaches as possible tools to limit MDR colonization in ICUs, there is no reference to current clinical evidence or recommendations regarding their use in cardiac surgery patients. The section on antibiotic cycling is overly brief and provides no practical conclusions.  While the authors correctly note the lack of long-term data in cardiac surgery, they omit broader evidence showing that antibiotic cycling has not demonstrated consistent efficacy in other settings either, and is no longer widely recommended. Similarly, the paragraph concerning “targeted prophylaxis and eradication” in MDR-colonized patients (e.g., CPE) is vague and potentially misleading. The authors do not clarify whether they are suggesting routine preoperative screening for MDR carriage and the use of targeted prophylaxis based on resistance profiles. Such an approach would contradict current guidelines (e.g., ESCMID/EUCIC 2023). This section should  be clarified.  

The phrase “for hospitalised patients or those with an LVAD” requires clarification. If the authors intend to highlight high-risk groups such as patients with prolonged hospitalization, MDR colonization, or LVADs, this should be stated explicitly. Otherwise, the reader may interpret that hospitalization itself (applying to nearly all cardiac surgery patients) is an indication for expanded antibiotic prophylaxis, which is inconsistent with current guidelines.

The phrase “pre-emptive viral load monitoring” is imprecise since  “pre-emptive” refers to a therapeutic strategy, not a diagnostic one. A more appropriate formulation would be “viral load monitoring with pre-emptive therapy.” Additionally, the sentence “combining antiviral prophylaxis with pre-emptive viral load monitoring” may be misleading. Antiviral prophylaxis and pre-emptive therapy are usually employed as alternative strategies depending on risk level and drug tolerance. Their combination as a so-called hybrid strategy (e.g., post-prophylaxis monitoring) is rare and is not the standard approach according to most guidelines (e.g., ISHLT, ESCMID). The authors should clarify in which population and based on what rationale such an approach would be justified, or reformulate the sentence in accordance with current evidence and recommendations.

The claim that “systematic use of targeted prophylaxis involving TMP-SMX and atovaquone” significantly reduced the incidence of opportunistic infections is also unclear. It should be clearly stated that TMP-SMX remains the first-line agent for prophylaxis against Pneumocystis jirovecii, Toxoplasma gondii, and Nocardia spp., while atovaquone is only used as an alternative in cases of TMP-SMX intolerance. The current phrasing may give the misleading impression that both agents should be used simultaneously.

The table requires a title and should be supplemented with essential clinical information. Dosing details (dose, timing of administration, intra- and postoperative redosing intervals) should be provided for all listed antibiotics. The rationale for combining cefazolin with piperacillin-tazobactam and vancomycin in heart transplantation  is unclear - what is the justification for such a combination, and what is its added value, given that the spectrum of piperacillin-tazobactam combined with vancomycin fully covers  that of cefazolin? Additionally, the MDR colonization section raises similar concerns as mentioned above.

References.  Many statements in the text are not adequately supported by citations. For example, in the sentence: “While few studies have found an increased risk of postoperative infections in cardiac surgery patients who were preoperative colonized by MDR pathogens…”, the authors do not cite any sources to support the existence of these “few studies.” Moreover, the reference section is disorganized, making it difficult to follow. The first citations in the text are numbered 4 and 19. Some references are separated by commas (e.g., 4,19), while others are formatted with dashes (e.g., 6-9-18), which may be confusing - is this a range or a formatting error? References 9 and 12 appear to be duplicates, and reference 46 is listed in the bibliography but not cited anywhere in the text. The entire reference list requires careful technical editing and consistent formatting.

Finally, although the authors mention varying levels of evidence throughout the manuscript, the overall structure would benefit from a more explicit differentiation between well-established, evidence-based recommendations, controversial areas, and knowledge gaps. A clearer distinction between these categories would enhance the practical utility of the review for clinicians. 

Comments on the Quality of English Language

Although the manuscript is generally understandable, the English language requires thorough editing.

Author Response

All comments have been addressed, inaccuracies have been corrected, and additional specifications have been added. The English text has been completely revised. A new article is attached.

Reviewer 3 Report

Comments and Suggestions for Authors

This comprehensive narrative review addresses a clinically important topic. Nevertheless, several methodological and content-related issues should be addressed.
1.The review lacks a clearly described search strategy. Please add a dedicated “Methods” subsection that states: databases searched , date range, MeSH and free-text keywords, inclusion/exclusion criteria.
2.Current recommendations rely heavily on North-American and European guidelines. Incorporate recent epidemiological data from Asia-Pacific, Latin America or Middle-Eastern ICUs to highlight regional differences in MRSA, CRE and Candida spp. prevalence and their impact on empirical choices.
3.The section on heart transplantation omits donor-derived infections (DDI). Please add a paragraph on pre-transplant donor screening  and targeted prophylaxis of multi-drug resistant Gram-negative bacilli or atypical mycobacteria.
4.After the 24–48 h prophylaxis window, include a flowchart for de-escalation/escalation decisions based on fever, PCT, cultures and imaging.
5.Add a supplementary table for pediatric, pregnant and renal-replacement patients: weight-based cefazolin dosing, vancomycin AUC/MIC targets, and safety considerations.
6.Although generally readable, the manuscript would benefit from professional English editing to improve conciseness and avoid minor grammatical errors.

Author Response

All comments have been addressed, inaccuracies have been corrected, and additional specifications have been added. The English text has been completely revised. The review only concerns adult patients, so paediatric patients are not covered. The available literature on pregnant women is too limited to allow for an in-depth review. A new article is attached.

Reviewer 4 Report

Comments and Suggestions for Authors

Thanks for giving me an opportunity to review. This narrative review explains the role of antibiotics when performing prophylaxis in cardiac surgeries. May I suggest some methods to make it better:

  • Can you make a separate table for pediatric cardiac conditions?
  • Are separate medications used for pediatric cardiac conditions?
  • Are there any differences by regions of the world? Are there differences by developed/ developing regions?
  • Are there any recommendations for pregnant and breastfeeding women?

Author Response

(The authors gave the same response as above.)

Round 2

Reviewer 2 Report

Comments and Suggestions for Authors

The current version of the manuscript has been significantly improved, but several issues still require clarification or modification.

A substantial number of statements throughout the manuscript are not supported by references:

There is not a single citation in the entire Introduction.

Outside the Introduction, the following fragments also require appropriate referencing:

“For cardiac surgical patients, the development of ventilator-associated pneumonia, bloodstream infection or sternal wound infection caused by MDR organisms can lead to prolonged ICU stays, higher costs and increased mortality.”

“One promising approach is the use of synergistic combinations of drugs or plant-derived compounds, such as ethyl gallate combined with tetracycline or fusidic acid, which have shown enhanced activity against MRSA.”

“Meanwhile, new β-lactam/β-lactamase inhibitor combinations, such as ceftolozane/tazobactam, have demonstrated efficacy against resistant strains of P. aeruginosa.”

Besides, ceftolozane/tazobactam is not the only new β-lactam/β-lactamase inhibitor with activity against MDR pathogens. It is unclear why the authors focus exclusively on this agent.

“The development of next-generation glycopeptides with enhanced potency is also progressing, particularly for treating vancomycin-resistant Enterococcus (VRE).”

“Other emerging strategies include the use of antimicrobial polymeric biomaterials and nanoparticle-based drug delivery systems, which offer targeted, sustained antimicrobial release, particularly in biofilm environments.”

“Furthermore, botanical extracts and phytochemicals with broad-spectrum activity are being explored, although most evidence remains at the preclinical stage.”

“In the meantime, alternative strategies such as antibiotic cycling, combination therapies, nanotechnology, phage therapy and probiotics can be used as valuable interim tools against critical MDR threats. Implementing these measures early and sustainably in high-risk units and surgical services can reduce transmission and improve clinical outcomes.”

This last passage raises particular concern - the authors suggest that implementing alternative strategies is beneficial, but the statement is vague and lacks clarity. It is not clear which specific intervention, how it is applied, and in which patient population might lead to transmission reduction or improved outcomes. Are there any data showing that any of the listed strategies (which one specifically and under what protocol) are clinically effective or prevent MDR transmission?

“A study at Stanford University found that the combination of inhaled amphotericin B and oral itraconazole for three months after transplant significantly reduced Aspergillus spp. infections, which are associated with high mortality. Additionally, HEPA-filtered respiratory masks have been found to be an effective non-pharmacological measure.”

“Nevertheless, surveys reveal that up to 74% of centres still adopt empirical prophylaxis, particularly for high-risk groups such as patients undergoing open-chest procedures.”

In addition, the definition of MDR remains incorrect and has not been revised. The authors should replace it with the correct definition and include the appropriate citation.

The following recommendation is also questionable:

“For high-risk patients (e.g. those undergoing reoperations or prosthetic implantation, or those with prior colonisation), consider administering a single preoperative dose of an aminoglycoside (e.g. gentamicin), Aztreonam, or a fluoroquinolone to broaden coverage.”

The implantation of a prosthetic device alone is not an indication for expanded prophylaxis. Cefazolin - unlike flucloxacillin or oxacillin used in the UK - provides some Gram-negative coverage (e.g., E. coli, Klebsiella). Furthermore, resistance among Enterobacterales to fluoroquinolones is high. It is unclear what the authors mean by "prior colonisation" - Enterobacterales are part of the normal gut microbiota in all humans. In addition, fluoroquinolones and aztreonam have limited activity  against ESBL- or carbapenemase-producing strains. The recommendation, as phrased, risks unjustified broadening of prophylaxis, which may not benefit the patient but increases the risk of resistance selection and microbiome disruption.

Author Response

Thank you for your review. Here are all the changes I have made:
- All bibliographical references have been added.
- The definition of MDR has been corrected.
- The statement on nanotechnology and phage therapy has been corrected, as there is still little evidence and it only caused confusion.
- For β-lactam/β-lactamase inhibitors that are active against multidrug-resistant pathogens, we have reported ceftolozane/tazobactam as one example, but we have also added another. We have not focused solely on ceftolozane/tazobactam.
- Patients at high risk of Gram-negative infections include those undergoing repeat surgery, especially if the procedure is complex or prolonged, as well as patients with prolonged preoperative hospitalisation/ICU, patients with LVAD in hospital and patients who are already colonised by MDR. This has been specified in the text.

Reviewer 3 Report

Comments and Suggestions for Authors

The author did not provide sufficient explanation or modification of the review‘s comments.

Author Response

Thank you for reviewing this. Here are all the changes that have been made.

1.The review lacks a clearly described search strategy. Please add a dedicated “Methods” subsection that states: databases searched , date range, MeSH and free-text keywords, inclusion/exclusion criteria.

corrected in the text

2.Current recommendations rely heavily on North-American and European guidelines. Incorporate recent epidemiological data from Asia-Pacific, Latin America or Middle-Eastern ICUs to highlight regional differences in MRSA, CRE and Candida spp. prevalence and their impact on empirical choices.

There is insufficient data available to enable us to draw reliable conclusions.

3.The section on heart transplantation omits donor-derived infections (DDI). Please add a paragraph on pre-transplant donor screening  and targeted prophylaxis of multi-drug resistant Gram-negative bacilli or atypical mycobacteria.

We did not find sufficient material regarding the donor's infection to be able to establish a solid basis.

4.After the 24–48 h prophylaxis window, include a flowchart for de-escalation/escalation decisions based on fever, PCT, cultures and imaging.  

For several reasons, descalation in these patients is not guided by inflammation index values:

Firstly, patients may have a fever 24 hours after surgery that is not related to an infection, and they are not treated for this.

Secondly, PCr and PCT values are altered by the inflammatory response that occurs with extracorporeal circulation, making them unreliable for diagnosing infection.

For these reasons, prophylactic antibiotics are discontinued without taking these parameters into account.

5.Add a supplementary table for pediatric, pregnant and renal-replacement patients: weight-based cefazolin dosing, vancomycin AUC/MIC targets, and safety considerations.

The article only concerns adult patients, which is why we have not added a section on paediatric patients.

6.Although generally readable, the manuscript would benefit from professional English editing to improve conciseness and avoid minor grammatical errors.

completely revised English

Round 3

Reviewer 2 Report

Comments and Suggestions for Authors

The paper has been substantially improved and is now suitable for publication.

Reviewer 3 Report

Comments and Suggestions for Authors

The author appears to have simply re-copied the initial reviewer response without implementing any of the suggested corrections. Given the lack of meaningful revisions, I recommend rejection of the manuscript.